# Apoptosis–Cell Cycle–Autophagy Molecular Mechanisms Network in Heterogeneous Aggressive Phenotype Prostate Hyperplasia Primary Cell Cultures Have a Prognostic Role

**DOI:** 10.3390/ijms25179329

**Published:** 2024-08-28

**Authors:** Elena Matei, Manuela Enciu, Mihai Cătălin Roșu, Felix Voinea, Anca Florentina Mitroi, Mariana Deacu, Gabriela Isabela Băltățescu, Antonela-Anca Nicolau, Anca Chisoi, Mariana Aşchie, Anita Cristina Ionescu (Mitu)

**Affiliations:** 1Center for Research and Development of the Morphological and Genetic Studies of Malignant Pathology, “Ovidius” University of Constanta, 145 Tomis Blvd., 900591 Constanta, Romania; elena_matei@365.univ-ovidius.ro (E.M.); mihai.rosu@365.univ-ovidius.ro (M.C.R.); anca.mitroi@365.univ-ovidius.ro (A.F.M.); gabriela.baltatescu@univ-ovidius.ro (G.I.B.); antonela.nicolau@365.univ-ovidius.ro (A.-A.N.); anca.dobre@365.univ-ovidius.ro (A.C.); 2Clinical Service of Pathology, “Sf. Apostol Andrei” Emergency County Hospital, 145 Tomis Blvd., 900591 Constanta, Romania; mariana.deacu@univ-ovidius.ro (M.D.); mariana.aschie@univ-ovidius.ro (M.A.); 3Medicine Faculty, “Ovidius” University of Constanta, 1 Universitatii Street, 900470 Constanta, Romania; felix.voinea@univ-ovidius.ro (F.V.); cristina-ionescu@365.univ-ovidius.ro (A.C.I.); 4Urology Department, “Sf. Apostol Andrei” Emergency County Hospital, 145 Tomis Blvd., 900591 Constanta, Romania; 5Chemical Carcinogenesis and Molecular Biology Laboratory, Institute of Oncology “Prof. Dr. Alexandru Trestioreanu”, 022328 Bucharest, Romania

**Keywords:** caspases 3/7 activity, cell cycle, aneuploidy status, adhesion glycoproteins, autophagy, nuclear shrinkage, oxidative stress, heterogeneous aggressive phenotype BHP cell cultures

## Abstract

Our study highlights the apoptosis, cell cycle, DNA ploidy, and autophagy molecular mechanisms network to identify prostate pathogenesis and its prognostic role. Caspase 3/7 expressions, cell cycle, adhesion glycoproteins, autophagy, nuclear shrinkage, and oxidative stress by flow-cytometry analysis are used to study the BPH microenvironment’s heterogeneity. A high late apoptosis expression by caspases 3/7 activity represents an unfavorable prognostic biomarker, a dependent predictor factor for cell adhesion, growth inhibition by arrest in the G2/M phase, and oxidative stress processes network. The heterogeneous aggressive phenotype prostate adenoma primary cell cultures present a high S-phase category (>12%), with an increased risk of death or recurrence due to aneuploid status presence, representing an unfavorable prognostic biomarker, a dependent predictor factor for caspase 3/7 activity (late apoptosis and necrosis), and cell growth inhibition (G2/M arrest)-linked mechanisms. Increased integrin levels in heterogenous BPH cultures suggest epithelial–mesenchymal transition (EMT) that maintains an aggressive phenotype by escaping cell apoptosis, leading to the cell proliferation necessary in prostate cancer (PCa) development. As predictor biomarkers, the biological mechanisms network involved in apoptosis, the cell cycle, and autophagy help to establish patient prognostic survival or target cancer therapy development.

## 1. Introduction

In prostate pathology cases, benign prostate hyperplasia (BPH), and prostate cancer (PCa), morphological changes in the normal prostate begin with inflammatory transition status. Inflammation and apoptosis mechanisms bring knowledge into the BPH pathogenesis, helping to understand the BPH etiology. Prostatic stroma presents various cells entrapped in an extracellular matrix, such as fibroblasts, smooth-muscle cells, endothelial cells, nerve cells, and T lymphocytes [1].

Apoptosis mechanisms, by cell surface receptors activation or cytochrome c release from mitochondria, determine the caspase cascades, which are divided into initiators (-2, -8, -9, -10, and -12) and effectors (-3, -6, and -7), with roles in cell cycle and apoptosis regulation [2]. Clinical studies have shown the role of inflammation in BPH in a study of 4000 patients by transurethral prostate resection (TURP) or prostatectomy, observing an acute or chronic inflammation in 43.1% of cases. The authors demonstrated a strong correlation between increased prostate volume and inflammation with neutrophil or mononuclear infiltrates [3].

A few in vitro studies showed apoptosis mechanisms in BPH cases. Epithelial cells in BPH overexpress the Bcl-2 level reported in healthy prostate tissue. Increased Bcl-2 expression shows an apoptotic mechanisms deregulation that promotes prostatic hyperplasia [4]. In the epithelial cells, caspase-3 expression in patients with BPH counteracts apoptosis, being implied in mitotic progression regulation. Many studies reported a strong correlation between inflammation, apoptosis, and PCa development. Inflammation and apoptosis processes are essential in BPH and PCa development, and several molecular pathways are known. But the factors implied in prostate carcinogenesis are mostly unknown [5].

Apoptosis, cell-cycle, and oxidative stress pathways in cancer cells represent critical targets in cancer therapy development [6]. Apoptosis represents a tumor growth control mechanism by counterbalancing prostate cell proliferation [7].

Oxidative stress is another important mechanism to study in prostate carcinogenesis because it determines cell-to-cell communication blockage, leading to decreasing cell proliferation and apoptosis [8,9]. Reactive oxygen species (ROS) releasing linked mechanism-DNA damage initiates tumorigenesis. In the functions of cancer cell phenotype and ROS level, there persists a dual role of cell proliferation. In various in vitro human cancer cell cultures, uncontrolled cell proliferation requires different signaling pathways, upregulation, and cell-cycle progression [10,11]. Interactions between tumor cells, platelets, and lymphocytes determine cell adhesion, leading to cancer progression and metastasis [12].

Autophagy is another interesting mechanism that inhibits tumor growth, limits inflammation, and eliminates the damaged mitochondria [13]. In oxidative stress conditions, autophagy determines damaged proteins and organelles accumulation [14,15,16], promoting tumor progression after cancer initiation [17,18,19,20].

Our study presents a major molecular mechanisms network in heterogeneous aggressive phenotype hyperplasia primary cell cultures reported to non-malignant cell cultures, highlighting the BPH pathogenesis and bringing new knowledge to prognostic establishment or targeting cancer therapy development.

Apoptosis deregulation by caspase-3/7 expressions by DEVD-MR/propidium iodide stain and oxidative stress by total reactive oxygen species count was analyzed by flow-cytometry methods. Changes in expressions of G0/G1, S, and G2/M phases of the cell cycle by PI stain highlight the aneuploidy status. Adhesion glycoprotein expression made by CD42b-PE stain reveals platelets and lymphocyte aggregation to tumoral and endothelial cells. The adaptive response to maintain cell survival under stress conditions was analyzed by studying autophagy and nuclear shrinkage mechanisms using Hoechst and acridine orange stain using flow-cytometry techniques. Furthermore, multiple regression was performed to determine the potential biomarkers with prognostic roles in patient survival in heterogeneous aggressive phenotype BHP cell cultures.

## 2. Results

### 2.1. Caspases 3/7 Activity

In heterogeneous BPH cell cultures reported to non-malignant adjacent prostate cell cultures (C1, negative control), the effector caspase-3/7 intracellular activity was analyzed by a flow-cytometry technique, being presented in Figure 1A–L.

The caspase-3/7 activation mechanism induces changes in cell viability, with significant differences between the samples and the control (S1: 8.61 ± 4.60; S4: 11.41 ± 1.85; S5: 10.55 ± 1.02 vs. C1: 86.95% ± 4.69, *p* < 0.05, Figure 1J–L).

The pro-apoptotic signal highlighted by the biochemical cascade shows significant changes in early apoptosis (EA) in the experimental prostate cell cultures reported to negative control (S1: 24.75 ± 16.50; S4: 10.45 ± 1.12; S5: 3.03 ± 1.10 vs. C1: 0.02 ± 0.01, *p* < 0.05, Figure 1J–L).

A particular interest is represented by late apoptosis, as a chronic response to injury induction and oxidative stress, which was highlighted by increased significant values in nodular hyperplasia, adenoma, and BHP cell cultures than the control (S4: 75.76 ± 1.20; S5: 72.70 ± 4.50; S1: 63.24 ± 10.65 vs. C1: 0.03 ± 0.01, *p* < 0.05, Figure 1J–L).

The BPH cell cultures (S1, S4, and S5) adopt modified apoptosis mechanisms with peaks of more than 10^6^ to 10^7^, as shown by the DEVD-MR/PI stain reported for non-malignant prostate cell cultures that present an apoptosis peak of 10^5^ to maximum 10^6^ (C1AP, Figure 1D–I).

### 2.2. Microenvironment Characterization of Nodular Hyperplasia and Adenoma Samples

The epithelial–mesenchymal transition of pro-inflammatory and profibrogenic phenotypes that characterize the prostate heterogeneous microenvironment of nodular hyperplasia and adenoma cell cultures is highlighted in Figure 2A–L.

Benign nodular hyperplasia sections used to establish the diagnosis of patients were characterized by acini proliferation with lobulated and nodular architecture. A dilated duct presence surrounded by the small and medium acini sizes was associated with stromal proliferation. Also, p63 and cytokeratin 34 (CK34 BE12) markers have positive reaction in basal cells from acini and Ki67 marker showed a negative expression that is correlated with benign lesion character (Figure 3A–D). Benign adenomatous hyperplasia samples studied in the diagnosed patient’s establishment were characterized by acini proliferation, interglandular distance presence, p63, and CK34 markers have positive reaction in basal acinar cells, and ki67 marker presented a negative expression in epithelial cells (Figure 4A–D).

### 2.3. Cell Cycle

As shown in Figure 5A–F,I by PI staining of the cell-cycle distribution and DNA ploidy, heterogeneous aggressive phenotype BHP cell cultures present cell growth inhibition by cell-cycle arrest in the G2/M phase (S1: 96.03 ± 0.23; S4: 90.99 ± 0.46; S5: 83.59 ± 0.94) reported to negative control (C1: 14.64 ± 2.44, *p* < 0.05, Figure 3I), characterizing the presence of the aneuploidy status.

The proliferative S phase revealed statistically significant increased values in prostate adenoma (S5: 14.83 ± 0.30) reported to non-malignant prostate-adjacent cell cultures (C1: 10.23 ± 0.35, *p* < 0.05, Figure 5C,F,H), being included in the high S-phase category (>12%) with a 50% higher risk of death or recurrence (worse prognostic) due aneuploid status presence rather than intermediary or low S-phase categories.

Nodular hyperplasia (S4) and BPH cell cultures (S1) showed statistically significant decreased values of the S phase of the cell-cycle reported to negative control, being included in the low S-phase category (<7%), and having a better prognostic value for the patient survival (S4: 6.16 ± 1.36; S1: 2.34 ± 0.57 vs. C1: 10.23 ± 0.35, *p* < 0.05, Figure 5A,B,D,E,H).

### 2.4. Cell Adhesion

A significant increase in CD42b transmembrane glycoprotein levels was observed in heterogeneous BPH cultures (S1: 48.21 ± 8.78; S4: 53.95 ± 6.98; S5: 67.73 ± 2.13) reported to negative control (C2: 26.24 ± 0.50, *p* < 0.05, Figure 6A–F).

Nodular prostate hyperplasia (S4) and adenoma heterogeneous primary cell cultures (S5) present increased CD42b-adhesion glycoprotein expressions reported for the BPH cell cultures and controls (Figure 6A–C), indicating platelets and lymphocytes recruiting to inflammation site due to the chronic inflammation presence in BPH patients. Increased integrin levels in BPH cell cultures suggest epithelial–mesenchymal transition (EMT), acquiring and maintaining an aggressive phenotype by escaping cell apoptosis and cell proliferation, representing a base for PCa development.

### 2.5. Autophagy and Nuclear Shrinkage

The heterogeneous BPH cell cultures highlighted in Figure 7A–E present pyknotic nuclei by Hoechst 33342 stain and lysosomal activity by acridine orange (AO) stain. In prostate heterogeneous phenotype adenoma cell cultures (S5) were observed statistically significantly higher values of H+ nuclear shrinkage and AO+ autophagy expressions than in the positive control (H+: 77.15 ± 3.74 vs. C1: 54.28 ± 16.12, *p* < 0.05; AO+: 92.08 ± 0.30 vs. C1: 76.13 ± 19.59, *p* < 0.05; Figure 7C,F,H,K,L).

In prostate nodular hyperplasia with aggressive phenotype heterogeneity (S4, atrophy, and chronic inflammation), cell cultures were observed a decreased significant statistical value of H+ nuclear shrinkage than the positive control (32.19 ± 0.98 vs. C1: 54.28 ± 16.12, *p* < 0.05, Figure 7B,I,K) and an increased AO+ autophagy expression than the positive control (AO+: 86.05 ± 0.11 vs. C1: 76.13 ± 19.59, *p* ≥ 0.05; Figure 7B,E,L).

A particular interest was represented by the BPH cell populations with double positively expressed by acridine orange and Hoechst dual stain (AO+H+). Most valuable observations were highlighted in the cell apoptosis program for the prostate primary heterogeneous adenoma cell culture (S5), being represented by increasing nuclear shrinkage and lysosomal activity reported to positive control (S5: 78.07 ± 3.41 vs. C1: 51.58 ± 17.41, *p* < 0.05, Figure 7C,J).

### 2.6. Oxidative Stress

Changes in the oxidative cellular stress mechanism in heterogeneous BPH cell cultures were analyzed by the total ROS flow-cytometry method, as presented in Figure 8A–G.

A linked apoptosis–oxidative stress mechanism was observed in primary BPH cell cultures with an aggressive phenotype, highlighting the lower ROS values reported for the positive control (S1: 32.50 ± 2.88 × 10^6^; S4: 35.50 ± 17.89 × 10^6^ vs. C1: 234.50 ± 224.92 × 10^6^, *p* ≥ 0.05, Figure 8A,B,D,E,G).

In prostate heterogeneous aggressive phenotype adenoma cell cultures (S5), the highest peak of ROS was observed and reported to control cell cultures (S5: 295.00 ± 17.32 × 10^6^ vs. C1: 234.50 ± 224.92 × 10^6^, *p* ≥ 0.05, Figure 8C,F,G).

### 2.7. Correlations between Apoptosis, Cell Cycle, Adhesion Glycoprotein Expression, Autophagy, and Nuclear Shrinkage in Different BPH Cell Cultures with Aggressive Phenotype

The relationships between molecular mechanisms in heterogeneous BPH cell cultures are presented in Figure 9A–D. Cell adhesion represented by CD42b+ glycoprotein expression was negatively correlated with the G0/G1 phase of the cell cycle (r = −0.57, *p* < 0.05) and viability (r = −0.69, *p* < 0.05) and positively correlated with immunoglobulin positive expression (IgG1+, r = 0.78, *p* < 0.01, Figure 9A).

Inhibition of cell growth represented by G2/M phase arrest presents negative strong correlations with viability (r = −0.86, *p* < 0.01) and the necrosis (r = −0.90, *p* < 0.01) mechanisms highlighted in heterogeneous BPH cell cultures with an aggressive phenotype (Figure 9B).

Cell proliferation, represented by the S proliferative phase of the cell cycle, was directly correlated with linked autophagy–nuclear shrinkage mechanisms (AO+H+, r = 0.78, *p* < 0.01) and necrosis (r = 0.85, *p* < 0.01), and negatively correlated with early apoptosis (r = −0.73, *p* < 0.01, Figure 9C).

Double positive BPH cell culture expression for lysosomal activity and nuclear shrinkage present a positive correlation with necrosis (r = 0.69, *p* < 0.01) and the S phase of the cell cycle (r = 0.78, *p* < 0.01, Figure 9D). 

### 2.8. Prognostic Role of Molecular Mechanisms Network in Heterogeneous BPH Cell Cultures

Furthermore, multivariate analysis regression analyzed the linked molecular mechanisms of late apoptosis and the S proliferative phase of cell-cycle expressions to observe their potential prognostic roles in heterogeneous aggressive phenotype BHP cell cultures (Figure 10A,B). Our analysis observed that late apoptosis and the S phase of the cell cycle represent unfavorable prognostic biomarkers, being dependent predictor factors for the molecular mechanisms network in heterogeneous aggressive phenotype BHP cell cultures.

## 3. Discussion

Our study analyzed cell death, DNA content, and the autophagy molecular mechanisms network by flow-cytometry methods to highlight prostate pathogenesis and its prognostic role in heterogeneous aggressive phenotype BPH primary cell cultures reported to non-malignant cell cultures.

Cell-death escape presents importance to target cancer therapies [21], involving molecular events by major apoptotic pathways, including the death receptor pathway (extrinsic) and the mitochondrial pathway (intrinsic) [22]. The essential caspases involved in cell-death mechanisms are initiators (caspase-2,-8,-9,-10) and effectors (caspase-3,-6,-7) [23,24,25].

Our study presented a pro-apoptotic signal highlighted by a caspase-3/7 cascade showing low changes in early apoptosis in heterogeneous BHP cell cultures. Increased late apoptosis by the caspases-3/7 pattern as a chronic response by injury induction and oxidative stress presents a particular interest in heterogeneous aggressive phenotype BPH cell cultures. This is because modified cells adopt modified apoptosis mechanisms with peaks of more than 10^6^ to 10^7^ by DEVD-MR/PI stain reported for non-malignant prostate primary cells with an apoptosis peak of 10^5^ to a maximum of 10^6^. The epithelial –mesenchymal transition by pro-inflammatory and profibrogenic phenotypes characterizes the prostate heterogeneous microenvironment of nodular hyperplasia and adenoma primary cell cultures.

In heterogeneous aggressive phenotype BHP cell cultures, a multivariate regression analysis showed that a higher late apoptosis status by caspases-3/7 cascade is an unfavorable prognostic biomarker, a dependent predictor factor for network molecular mechanisms by cell adhesion by CD42b+ glycoprotein expression, cell-cycle blockage by G2/M phase arrest, and oxidative stress by total ROS count. An apoptosis–oxidative stress mechanisms network was highlighted by the highest peak of ROS reported to control in prostate heterogeneous aggressive phenotype adenoma cell cultures.

A recent study that highlighted prostate cancer progression by cell proliferation and apoptosis imbalance reported a silent increase in early apoptosis by FITC/PI stain in PCa patients and a higher necrosis pattern in BPH patients than in healthy patients [26]. Apoptosis results after activation mechanisms of initiator and executioner caspases [27], leading to chromatin condensation, cell adhesion loss, cell shrinkage, membrane blebbing, and DNA fragmentation. Mutations of caspase-3 activity were identified in different human cancers, such as colon and stomach cancer, lymphomas, and hepato-carcinoma [28,29,30,31,32,33]. In breast and prostate cancers, caspase-3 activity was identified, but without mutations [31,32,33]. On the other hand, caspase-7 expression presents mutations in human cancers [33,34]. Lower caspase-7 activity was observed in gastric carcinoma cases, but caspase-3 showed a higher pattern than non-malignant mucosa samples [35,36]. In acute myelogenous leukemia (AML) patients were observed increased caspase-3 levels reported to patients with normal peripheral blood lymphocytes [37]. Caspase-3 activity, expressed in breast and prostate carcinoma, is positively correlated with tumor progression [28,29]. Macrophage infiltration in breast cancer cells determines tumor progression [38]. The associated macrophages of tumor progression were reported in human cancer [39], and cancer apoptotic cells represent a source of chemo-attractants by stimulating macrophage infiltration [40]. In a dormant tumor, the continuous disappearance of cells by apoptosis allows a proliferation degree [41], but accumulated mutations lead to increased cell malignancy [42]. In both apoptotic ways, tumor progression initiation depends on the functional caspases, suggesting a caspase-dependent mechanism to release chemo-attractant factors from apoptotic cells with roles in macrophage infiltration [43]. Errors in the signaling pathways of cellular differentiation stimulate the cell proliferative program, which depends on non-apoptotic caspase activities. In many reports, it was found that caspase activity is implied in cellular proliferation. Caspase-3 activity is directly correlated with the cell cycle in HeLa cells, having an inhibition peak in the G2/M stage [44]. Caspase-7 expressions showed activated mitosis in tumor cell lines, leading to mitotic arrest, suggesting their role in tumor proliferation [45]. Another study reported that caspase-3 activity lost in transgenic mice models induces pancreatic beta cell hyper-proliferation by the c-myc signaling pathway, showing their role in the cell cycle of tumor cells [46].

The modified cell apoptosis mechanism in tumorigenesis is fundamental because acquired apoptosis resistance during therapy represents a major obstacle to cancer treatment. Many reports showed that tumor cells adapt to evading apoptosis by losing caspase function in cancer cell cultures [37]. Biochemical cascades highlighted in apoptosis are represented by caspase-3, -6, and -7 activity. Cytochrome c, released from mitochondria and its subsequent binding to caspase-9, determines caspase-3 activation, which is the executioner caspase essential for the nuclear changes associated with apoptosis, including chromatin condensation. A central role of the biochemical caspase cascade, implied in immune response and prostate cancer cell apoptosis, was studied to caspase-1, -3, and -9 expressions in normal and malignant human prostate samples [32].

Various studies propose two different types of cell-death mechanisms: accidental cell death (ACD) and regulated cell death (RCD) [47,48]. Physical and chemical stress factors exceeding the cell’s ability to restore their functions determine ACD, characterized by morphological events such as cell membrane alterations and cellular content release [49]. RCD is regulated by specific signaling pathways that determine biochemical, morphological, and immunological consequences in cells. The most studied ACD mechanism based on cell morphology, gene expression, and biochemical properties is autophagy [50,51,52,53,54,55]. In the meantime, regulatory molecules from RCD pathways represent biomarkers that are useful as potential therapeutic targets [56]. PCa is the second maligned affection and the fifth global cause of cancer death in men [57]. In apoptosis, reactive oxygen species (ROS) production causes a pro-inflammatory environment. Apoptosis-inducing factor (AIF), Fas ligand (FasL), and TNF-related apoptosis-inducing ligand (TRAIL) are death-inducing biomarkers that act on FasL/TNF receptor (TNFR)1, inducing apoptosis dependent upon caspase activity [54]. Apoptosis is visualized in different tissue samples using acridine orange (AO) [58,59], highlighted in areas with lysosomal and phagocytosis activity. Another study showed that the necrotic-like cell phenotypes required by gene activation and protein synthesis are programmed cell-death forms [60] with morphological characteristics of necrosis and apoptosis [61,62]. Apoptosis directly correlates with the cell cycle, plasma membrane changes, chromatin condensation, and DNA fragmentation. Damage-related pattern molecules (DAMPs) and mitochondrial DNA are released into the extracellular environment. Apoptosis is characterized by limited DMAP release. Membrane death receptor activation via the TNF, TRAIL, and FasL pathways and intracellular stimulation via genetic damage, hypoxia, and oxidative stress trigger apoptosis [63]. Mitochondrial apoptosis induced by cytochrome c, SMAC/DIABLO, HrtA2/Omi, and AIF factors determines apoptosis. Cytotoxic factors activate the BH3 proteins that determine BAX and BAK activation with increasing mitochondrial permeability. Cytoplasmic cytochrome c interacts with apoptotic protease activating factor 1 (APAF1), leading to apoptosomes initiation that stimulates pro-caspase-9, -3, and -7 activities, determining the dysfunction of cellular components and apoptosis [54,58,64].

Because BPH heterogeneity limits treatment response, understanding the molecular mechanisms involved in patient prognostic survival is necessary. Our study presents cell cycle, aneuploidy status, and cell adhesion by flow-cytometric analysis to highlight the heterogeneously microenvironment with aggressive phenotype BPH primary cell cultures reported to non-malignant cell cultures. Cell growth inhibition by cell-cycle arrest in the G2/M phase characterizes the aneuploidy status of DNA. The proliferative S phase of the cell cycle in prostate adenoma cell cultures is included in the high S-phase category (>12%) with a higher risk of death or recurrence (worse prognostic) due to aneuploid status presence. Also, nodular hyperplasia and BPH primary cell cultures with S phases included in the low S-phase category (<7%) present a better prognostic for patient survival. Inhibition of cell growth represented by G2/M phase arrest presents negatively strong correlations with viability and necrosis. In the meantime, cell proliferation, represented by the S phase of the cell cycle, was positively correlated with autophagy–nuclear shrinkage and necrosis, and negatively correlated with early apoptosis in BPH cell cultures. In our multivariate regression analysis, the S proliferative phase represents an unfavorable prognostic biomarker, being a dependent predictor factor for the molecular mechanisms network represented by caspase-3/7 activity (late apoptosis and necrosis) and cell growth inhibition (G2/M arrest) in heterogeneous aggressive phenotype BHP cell cultures.

Aneuploidy status is a tumorigenesis driver, representing a prognostic biomarker of tumor progression [65,66]. Aneuploidy cells are found in various tumors, often indicating higher malignancy [67]. Aneuploidy flow-cytometric analysis was used as a prognostic indicator in prostate, colon, and breast tumors [68]. The authors reported that cell distribution in the G0/G1, S, and G2/M phases by flow cytometry represents a rapid and efficient method for prognostic analysis [69,70]. The S phase has a prognostic role in patients, being divided into low (<7.0%), intermediate (7.0–11.9%), and high (≥12%) categories. The risk of death or recurrence for aneuploid cases is increased for the high S-phase category than for the intermediary or low categories. Different techniques or samples do not affect the cell cycle because a high S phase remains correlated with a worse tumor grade in breast cancer tissue samples [71].

The regulatory proteins implied in cell cycle guarantees damaged DNA elimination from cells to assure the homeostasis. Cell-cycle regulators represent targets for cancer therapies [72]. Cell-cycle regulation is guaranteed by the retinoblastoma tumor suppressor protein (Rb), p53, p27, or p21 being described in cancer [73,74,75]. In the cell cycle, Rb regulates the G1 to S phase’s transition [76]. In oxidative stress and cell damage, the p53 protein (TP53 gene) represents the control point as a tumor suppressor and pleiotropic transcription factor determining cell-cycle arrest. When cells do not repair DNA damage, p53 induces an apoptosis process mediated by the Bcl-2 family [77]. DNA damage causes the p21 and p53 response, determining the cell-cycle arrest in the G1 and G2 phases [78]. In phase S, p21 inhibits the cyclin A-CDK1 complex by blocking the transition to the G2M phase [79]. The p21 protein represents the independent anti-proliferative effector reported for the p53 pathway [80]. A higher p21 level induces prostate cancer aggressiveness. In the meantime, its lower values act against prostate tumorigenesis [81]. The p27 protein is a member of the CIP/KIP family. P27 regulates the G1 to S phase transition by cyclin-dependent kinases (CDKs) inhibition [82]. P27 regulates apoptosis. A lower p27 level is associated with aggressive tumor behavior [83,84].

Our study presents increased CD42b+ transmembrane glycoprotein levels in prostate hyperplasia heterogeneous primary cell cultures, indicating platelets and lymphocytes recruiting to the inflammation site due to the chronic inflammation presence in BPH patients. Increased integrin levels in heterogeneous BPH cultures suggest epithelial–mesenchymal transition (EMT), acquiring and maintaining an aggressive phenotype by escaping cell apoptosis and cell proliferation, leading to PCa development. Cell adhesion represented by CD42b+ glycoprotein expression was negatively correlated with the G0/G1 phase of the cell cycle and viability and positively correlated with immunoglobulin-positive expression (IgG1+). Integrins have an essential role in cancer progression. Modified cell adhesion biomarkers lead to cell proliferation, migration, and metastasis, being correlated with tumoral stages and outcomes (recurrence and survival) [85,86,87,88,89,90]. In another study, the CD42b+ adhesion glycoprotein pattern was increased in PCa cases [26]. Other reports presented CD42b glycoproteins as having essential roles in cell signaling networks, growth, differentiation, and survival [91,92,93].

Autophagy, another essential mechanism in our study, was presented in heterogeneous aggressive phenotype BHP cell cultures by a double positive expressed by acridine orange and Hoechst stain (AO+H+). Significant changes in the apoptosis program by increasing nuclear shrinkage and lysosomal activity reported to non-malignant cell cultures were observed in heterogeneous primary prostate adenoma cell cultures. Double positive heterogeneous BPH cell culture expression for lysosomal activity and nuclear shrinkage presents a positive correlation with necrosis and the proliferative S phase of the cell cycle.

Autophagy transports damaged proteins and organelles at the lysosomes for degradation. In prostate cancer cells, when the mitochondrial dysfunction is present, cytochrome c is released from the mitochondria in the cytoplasm, determining the caspase-3 activation and cell apoptosis. ROS generation activates the cell-death processes [47,94,95]. Autophagy promotes cellular adaptation to survival in stressful conditions [96]. The anticancer factors induce autophagy in different cancer cells [97,98]. GnRH-II antagonist Trp-1 determines the inhibition of cell proliferation and death signaling pathway activation in human prostate cancer cell cultures by mitochondrial dysfunction and autophagy [99,100]. Autophagy presents autophagosome formation, fusion with the lysosome, autophagy body breakdown, and recycling. Similar settings were observed between autophagy and apoptosis pathways [101]. The difference between apoptotic pathways depends on the physiologic status, developmental stage, and tissue type [102].

Mitochondria represent the principal organelles that play a role in apoptosis and autophagy [103]. Akt (protein kinase B) and p70S6 kinase regulation involve similar signaling pathways in apoptosis and autophagy. The cell death–autophagy network includes DAPk, Beclin 1, and protymosin-α [104]. Genetic alterations in the cell proliferation/apoptosis regulation pathways in cancer development suggest an autophagy–apoptosis network [103]. In tumor progression, autophagy pathway deregulation determines autophagy activity reduction, highlighted in its function as a safeguard mechanism in uncontrolled cell growth restriction, a protective mechanism against apoptosis.

An autophagy vesicle’s presence in dying cells reflects an adaptive response to maintain cell survival under stress conditions. In cancer cells, apoptosis and autophagy pathways depend on response severity, cell constituents, and other signaling pathways. Apoptosis and autophagy are not mutually exclusive, but various morphologic changes are still attributed, in part, to distinct biochemical and molecular events [103,104]. In certain cells, autophagy morphological features occur prior to apoptosis. Controversy exists when autophagy is constantly increased because it is necessary to elucidate the molecular basis in cell death by studying new biomarkers. BPH and cancer influence homeostasis, assuring an unstable report between proliferation and cell death [105]. Significant changes in the mitosis–apoptosis balance affect homeostasis, exposing patients to immune disorders, degenerative diseases, and cancer [106].

Future directions in the research area highlight the central molecular mechanisms network involved in apoptosis, the cell cycle, and autophagy as predictor biomarkers useful for establishing patient prognostic survival in different malignant affections or targeting cancer therapy development.

## 4. Materials and Methods

### 4.1. Materials

Various tissue samples excised by transurethral resection of the prostate (TURP), recovered from patients with benign prostatic hyperplasia (BPH, who signed an informed consent form, agreeing to participate in this study, n = 12) from the Clinical Service of Pathology, Sf. Apostol Andrei Clinical Emergency County Hospital in Constanta, Romania, were mechanically homogenized with TissueRuptor II (Qiagen, Hilden, Germany).

### 4.2. Prostate Primary Cell Cultures

Different cells and small tissue fragments are placed in a tissue-culture flask with a growth area (25 cm^2^) cultured in DMEM High Glucose and mixed in humidity conditions of 5% CO_2_ at 37 °C for 14 days. Prostate cells were distributed in tissue-culture test plate 6 and growth-enhanced treated (ThermoFisher Scientific Inc., Waltham, MA, USA). After 14 days, caspase-3/7 activity, cell cycle, cell adhesion, nuclear shrinkage, lysosomal activity, and oxidative stress were performed by flow-cytometry methods in the Cell Biology Department, CEDMOG, Ovidius University of Constanta, Romania.

### 4.3. Samples and Controls

Experimental heterogeneous BPH primary cell cultures with an aggressive phenotype were divided in three types of groups: 1-BPH (S1–S3; S6–S12); 2-nodular BPH, atrophy, and chronic inflammation (S4); 3-prostate adenoma with chronic inflammation (S5), being studied the biological processes network implied in tumorigenesis. Experimental BPH primary cell cultures were reported to negative and positive controls. The negative control is represented by prostate primary cell cultures developed from non-malignant adjacent tissue samples recovered from diagnosed patients with BPH. The positive control is represented by prostate hyperplasia primary cell cultures developed from prostatic tissue samples recovered from diagnosed patients with BPH.

### 4.4. Nodular Hyperplasia and Adenoma Microenvironment Characterization by IHC Methods

After the macroscopic description, prostate tissue samples were fixed in 10% formaldehyde, paraffin-embedded, sectioned, and stained in hematoxylin–eosin (HE) [26]. Monoclonal antibodies from immunohistochemical methods (IHC) were applied to selected sections of the study case. The antibody anti-Ki67 (Master Diagnostica, clone SP6, Sao Paolo, Brazil) is used to evaluate nuclear expression. HMWCK (clone 34BE12) and p63 (Master Diagnostica, clone 4A4, Sao Paolo, Brazil) are used to observe basal cell layer expressions, being basal cell markers. The basal cell layer is present in the histologically normal prostate and in benign lesions, while it is absent in prostatic carcinoma. Also, Ki67 expression represents a nuclear proliferation marker, being negative in benign lesions (BPH) and positive in prostate cancer (PCa).

### 4.5. Equipment

Our analysis was performed on an Attune Acoustic focusing cytometer (Applied Biosystems, Waltham, MA, USA). The flow cytometer was set using Attune performance-tracking beads, labeling, and detection (Life Technologies, Europe BV, Bleiswijk, The Netherlands) [107]. More than 10,000 cells per sample for each analysis were gated using the flow cytometer’s forward scatter (FSC) and side scatter (SSC). Data were collected and interpreted using Attune Cytometric Software v.1.2.5, Applied Biosystems, 2010. Heterogeneous prostate hyperplasia cell cultures with an aggressive phenotype (nodular hyperplasia with chronic inflammation and adenoma with chronic inflammation) were analyzed using a Primo Star optical microscope (Zeiss, Gottingen, Germany).

### 4.6. Methods

#### 4.6.1. Caspase-3/7 Activity

Cell apoptosis was observed using the DEVD-MR/PI methodology (FAM Caspase-3/7 Assay Kit, Abcam, Boston, MA, USA). Prostate primary cell cultures (BPH cells or non-malignant cells-negative control-C1, 200 μL) were transferred in tubes, and 20 μL of DEVD-MR solution and 20 μL of PI, were added, mixed, and incubated for 30 min at room temperature in darkness. Also, an intern control tub with experimental unstained BPH cells (NS) was made. One mL FCB (eBioscienceTM, Life Technologies Europe BV, The Netherlands) is added. Viability, early and late apoptosis, and necrosis were analyzed by BL3 channel (DEVD-MR) and BL2 channel (PI) [108].

#### 4.6.2. Cell Cycle

Two hundred µL of prostate primary cell cultures (BPH cells or non-malignant cells-negative control-C1) were fixed with 200 µL absolute ethanol for 30 min. Then, the fixed prostate cells were treated with 10 µL of PI (20 mg/mL) and kept in darkness at room temperature for 30 min. After adding 1 mL FCB, the cell-cycle phases were analyzed using the BL2 channel (PI) [109].

#### 4.6.3. CD42b Adhesion Glycoprotein

Anti-CD42b-PE (HIP1, Invitrogen, eBioscience, Life Technologies Corp, Carlsbad, CA, USA) monoclonal antibodies conjugated with phycoerythrin (PE), were used to assess the GPIba platelet glycoprotein in different experimental samples. Prostate cell cultures (100 µL) were spread for each tube, including 1.—experimental CD42b-PE stain; 2.—positive control-IgG 1 stain; 3.—negative controls (C2-prostate non-malignant cells and NS-unstained BPH cells). In experimental tubes with BPH cells, 5 µL of CD42b-PE were added. A positive control tube, with BPH cells and 5 µL mouse IgG 1, was made. Another two tubes as negative controls were prepared, one tube with prostate non-malignant cell cultures (C2) with 5 µL of CD42b-PE and another tube with unstained BPH cells (NS) to report the experimental tubes. All tubes were vortexed and incubated in darkness for 25 min at 37 °C. One ml of flow-cytometry stain buffer (FCB) was added and vortexed before analysis. To identify the CD42b adhesion glycoprotein by size and specificity to flow cytometer was used the BL2 channel (PE) [26].

#### 4.6.4. Nuclear Shrinkage and Lysosomal Activity

In primary prostate cell cultures (BPH cells or positive control, 300 μL), two μL of Hoechst 33342 stain, and 50 μL of acridine orange (1.0 μM, AO) were added into the tubes, mixed, and incubated in darkness at room temperature for 30 min. Also, an intern control tub with experimental unstained BPH cells (NS) was made. Cells were analyzed by flow cytometry after 0.5 mL FCB addition using UV excitation to the VL2 channel (Hoechst) and BL1 channel (AO) [109].

#### 4.6.5. Total Reactive Oxygen Species (ROS)

The total reactive oxygen species (ROS Assay Kit 520 nm, Invitrogen, USA) was used to identify the linked cell signaling pathway. After 60 min at 37 °C and 5% CO_2_ incubation, stained cells (1 mL of BPH cells or positive control) with two μL 500X ROS stock solution were analyzed in the BL-1 channel. Also, an intern control tube with experimental unstained BPH cells (NS) was made [110].

### 4.7. Data Analysis

Our results were presented as means ± standard deviation (SD), representing flow-cytometry methods as caspase-3/7 activity (%), cell cycle (%), nuclear shrinkage (%), autophagy (%), adhesion glycoprotein (%, integrin CD42b), and oxidative stress count (×10^6^). To identify the normal distribution of parameters, the Kolmogorov–Smirnov test was used. The Mann–Whitney test was applied to establish the differences between samples and controls, and *p* < 0.05 was considered statistically significant, made by MedCalc v20.111 Software Ltd. (Ostend, Belgium). Parametrical correlations (Pearson test, r) between apoptosis, cell cycle, adhesion glycoprotein expression, autophagy, and nuclear shrinkage in different BPH cell cultures were analyzed. Furthermore, multiple regression was performed by MedCalc v20.111 Software Ltd. (Ostend, Belgium) to determine the potential prognostic values of the linked mechanisms of late apoptosis and the S proliferative phase of the cell cycle in heterogeneous phenotypically BHP cell cultures. Figure 1A–I, Figure 5A–F, Figure 6A–C, Figure 7A–H and Figure 8A–F were made with Attune Cytometric Software v.1.2.5, Applied Biosystems, 2010 (Bedford, MA, USA) and Figure 1J–L, Figure 5G–I, Figure 6D–F, Figure 7J–L, Figure 8G, Figure 9A–D and Figure 10A,B with MedCalc v20.111 Software Ltd. (Ostend, Belgium).

## 5. Conclusions

Different factors limit a treatment response in aggressive phenotype BHP patients, so studying and understanding the molecular mechanisms involved in prognostic survival is necessary. Our study observed that a high late apoptosis expression, a high proliferative S phase of the cell-cycle category, and aneuploidy status represent unfavorable prognostic biomarkers and dependent predictor factors for the major biological mechanisms network in heterogeneous aggressive phenotype BHP primary cell cultures.

## Figures and Tables

**Figure 1 ijms-25-09329-f001:**
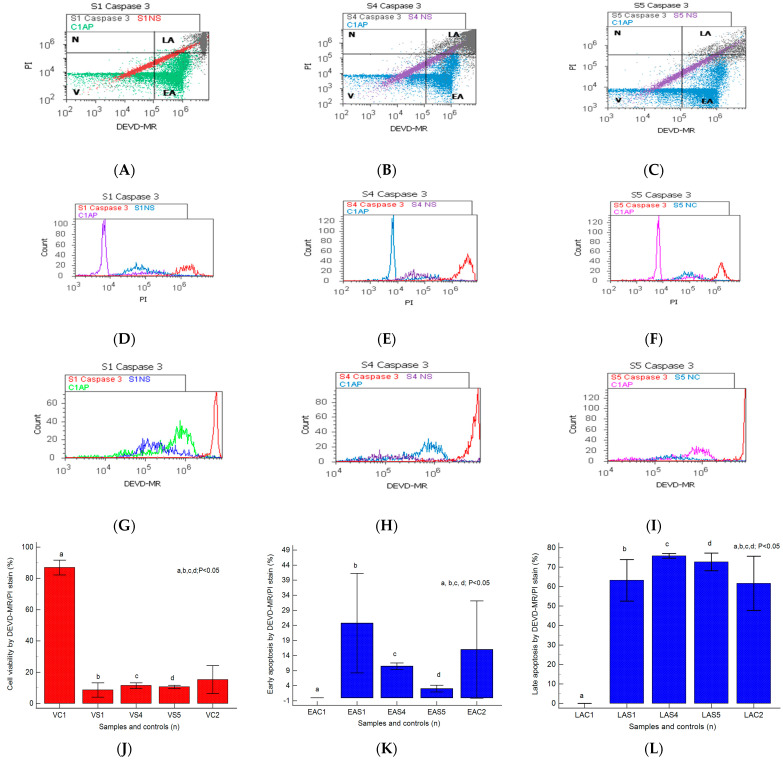
Caspases-3/7 activity plotted by DEVD-MR/PI stain highlighted to observe viability (V), early and late apoptosis (EA; LA), and necrosis (N) in heterogeneous BPH cell cultures (S1, S4, and S5) reported to non-malignant prostate cell cultures (C1AP). (**A**–**C**) V: A—15.48%; B—9.81%; C—9.66%; C1—82.88%; EA: A—40.62%; B—11.42%; C—3.99%; LA: A—42.94; B—76.80%; C—76.61%; N: A—0.94%; B—1.95%; C—9.82%; C1—17.12%. (**D**–**I**) Cell apoptosis escape pathway in heterogeneous BPH cell cultures by PI stain (**D**–**F**): 20 × 10^6^ (S1); 45 × 10^6^ (S4); 40 × 10^6^ (S5) vs. 12.5 × 10^5^ (C1AP); by DEVD-MR stain (**G**–**I**): 65 × 10^7^ (S1); 90 × 10^7^ (S4); 130 × 10^7^ (S5) vs. 35 × 10^6^ (C1AP). (**J**–**L**): Caspases 3/7 expression statistics of BPH patients reported to controls (S1; S4; S5; C1; C2); a, b, c, d; *p* < 0.05 represent significant statistical differences between samples and controls (Mann–Whitney test by MedCalc program). Legend: A—benign prostatic hyperplasia (S1); B—prostate nodular hyperplasia, atrophy, and chronic inflammation (S4); C—prostate adenoma with chronic inflammation (S5); NC—unstained control prostate cells (S1, S4, and S5); C1AP—non-malignant adjacent prostate cells (C1, negative control); C2—positive control (BPH).

**Figure 2 ijms-25-09329-f002:**
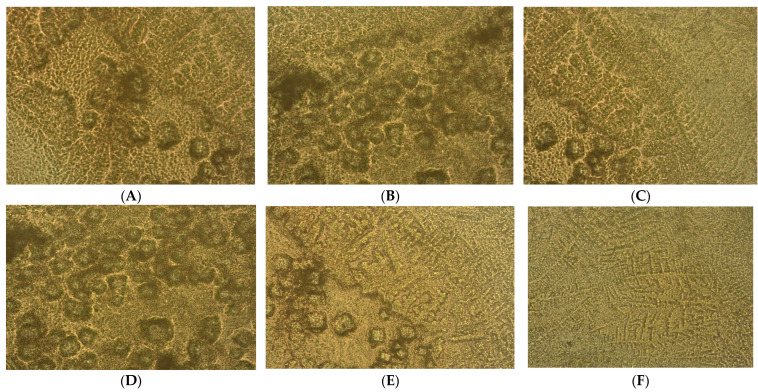
Heterogeneous primary prostate cell cultures with aggressive phenotype developed from tissue fragments recovered from diagnosed patients with benign prostate hyperplasia (BPH) after 14 days in vitro development by optical microscopy (×40). (**A**–**F**) Prostate nodular hyperplasia, atrophy, and chronic inflammation cell cultures (S4) highlighting acini presence with nodular architecture and stromal proliferation; (**G**–**L**) prostatic epithelial cells proliferation from adenoma with chronic inflammation cell cultures (S5).

**Figure 3 ijms-25-09329-f003:**
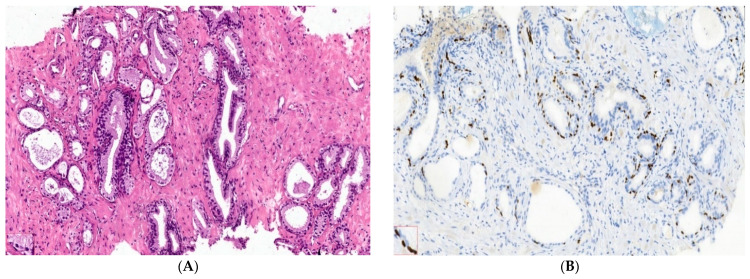
IHC staining methods by optical microscopy were used to establish the benign prostatic hyperplasia diagnosis in patients. Nodular hyperplasia, atrophy, and chronic inflammation (**A**–**D**) tissue-fragment samples were used to develop in vitro heterogeneous primary prostate cell cultures with an aggressive phenotype (S4). Legend: A—benign prostatic hyperplasia acini proliferation with lobulated, nodular architecture, dilated duct presence surrounded by the small and medium acini size, associated with stromal proliferation (HE; ×100); B—benign prostatic hyperplasia p63—marker—positive in basal cells in evaluated prostatic acini (×100); C—HMWCK (clone 34BE12)—continuously positive in basal cells in acini with lobulated architecture (×100); D—Ki67 marker-negative correlated with benign lesion character (×100).

**Figure 4 ijms-25-09329-f004:**
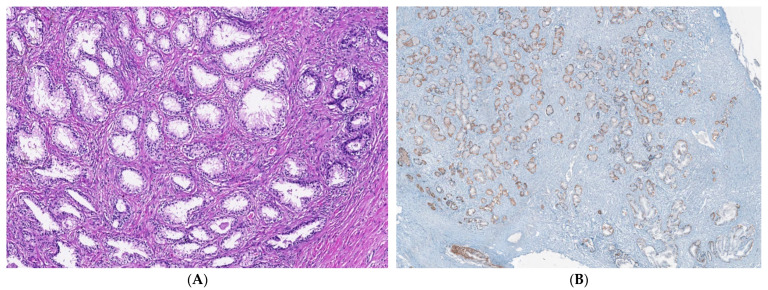
IHC staining methods by optical microscopy were used to establish benign adenomatous hyperplasia diagnosis in patients (**A**–**D**). Prostate adenoma with chronic inflammation tissue-fragment samples were used to develop in vitro heterogeneous primary prostate cell cultures with an aggressive phenotype (S5). Legend: A—benign adenomatous hyperplasia, with prostatic acini proliferation and interglandular distance presence (HE; ×100); B—p63 marker—continuously positive in basal acinar cells (×40); C—HMWCK (clone 34BE12)—continuously positive in basal acinar cells (×100); D—Adenomatous prostatic hyperplasia—ki67 marker-negative in prostatic epithelial cells (×100).

**Figure 5 ijms-25-09329-f005:**
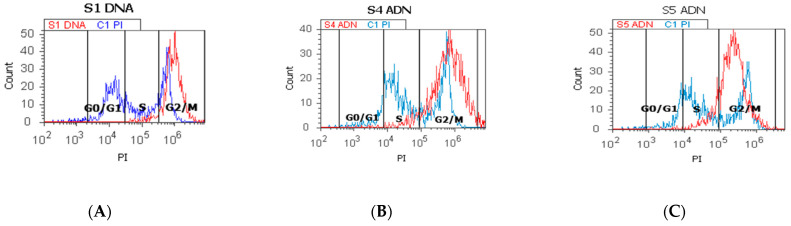
Cell cycle represented by propidium iodide stain (PI). (**A**–**C**): control DNA (C1) is extrapolated on the PI ax; G0/G1 phase = A—0.18%; B—0.44%; C—0.45; C1—67.70%; S phase = A—4.80%; B—4.98%; C—14.57%; C1—10.54%; G2/M phase = A—94.27%; B—90.54%; C—82.78%; C1—12.53%. (**D**–**F**): Arrest of G2/M phase in heterogeneous BPH cells (S1, S4, and S5) reported to non-malignant prostate cells (C1PI). (**G**–**I**): Cell-cycle statistics of BPH patients reported to controls (S1; S4; S5; C1; C2); a, b, c, d; *p* < 0.05 represent significant statistical differences between samples and controls (Mann-Whitney test by MedCalc program). Legend: A; D—benign prostatic hyperplasia (S1); B; E—nodular hyperplasia, atrophy, and chronic inflammation (S4); C; F—prostate adenoma with chronic inflammation (S5); C1PI—non-malignant adjacent prostate cells (negative control); C2—positive control (BPH).

**Figure 6 ijms-25-09329-f006:**
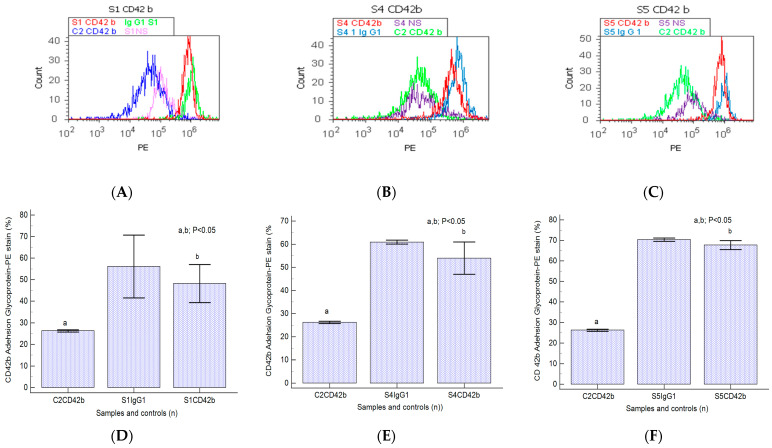
Integrin expression in platelet aggregation to tumoral cells by highlighting adhesion glycoproteins with CD42b-PE stain in heterogeneous BPH cells (**A**–**C**) reported to non-malignant prostate cells (C2). (**A**–**C**): CD42b+ expression: A—40.60%; B—47.90%; C—65.88%; C2—25.80%. (**D**–**F**): CD42b glycoprotein expressions statistics of BPH patients reported to controls (S1; S4; S5; C2; Ig-G1); a, b; *p* < 0.05 represent significant statistical differences between samples and controls (Mann–Whitney test by MedCalc program). Legend: A—benign prostatic hyperplasia (S1); B—prostate nodular hyperplasia, atrophy, and chronic inflammation (S4); C—prostate adenoma with chronic inflammation (S5); NS—unstained control prostate cell cultures (S1, S4, and S5); C2—non-malignant adjacent prostate cell cultures (negative control); IgG1—positive control prostate cell cultures (S1, S4, and S5).

**Figure 7 ijms-25-09329-f007:**
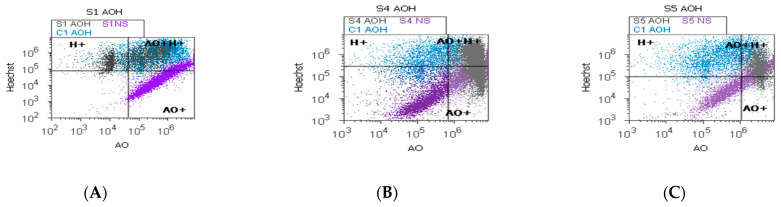
Nuclear shrinkage and autophagy plotted by Hoechst–acridine orange (AO) dual stain. (**A**–**C**): AO^+^H^+^: A—38.28%; B—44.43%; C—81.03%; C1—48.18%; H^+^: A—60.47%; B—31.34%; C—80.40%; C1—61.79%; AO^+^: A—42.08%; B—85.95%; C—91.82%; C1—37.09%. Lysosomal activity (**D**–**F**) and nuclear membrane integrity (**G**–**I**) changes in heterogeneous BPH cell cultures (S1, S4, and S5) reported to positive control prostate cell cultures (C1) and unstained control prostate cell cultures (S1NS; S4NS; S5NS); (**J**–**L**) nuclear shrinkage and autophagy statistics of BPH patients reported to controls (S1; S4; S5; C1); a, b, c; *p* < 0.05 represent significant statistical differences between samples and control (Mann–Whitney test by MedCalc program). Legend: A; D; G—benign prostatic hyperplasia (S1); B; E; H—nodular hyperplasia, atrophy, and chronic inflammation (S4); C; F; I—prostate adenoma with chronic inflammation (S5); NC—unstained control prostate cell cultures (S1, S4, and S5); C1—positive control prostate cell cultures.

**Figure 8 ijms-25-09329-f008:**
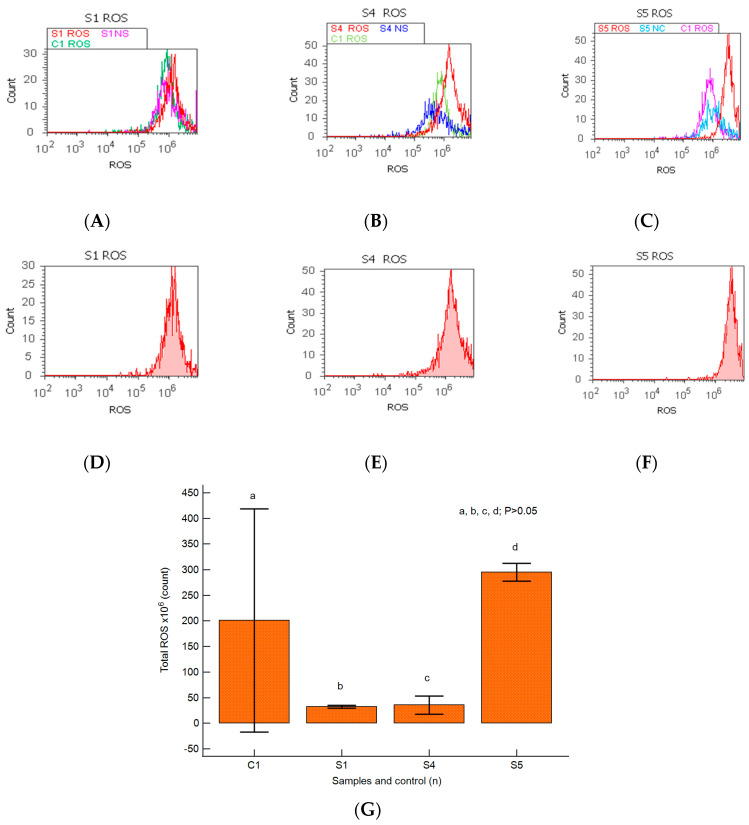
Total reactive oxygen species pattern (ROS). (**A**–**F**) Oxidative stress changes in heterogeneous BPH cells (S1, S4, and S5). ROS: D-30 × 10^6^; E-51 × 10^6^; F-28 × 10^7^; C1-35 × 10^6^. (**A**–**C**) Positive control prostate cells (C1ROS) and unstained control prostate cells (S1NS, S4NS, and S5NS) are extrapolated on the ROS ax. (**G**) Oxidative stress pattern statistics of BPH patients reported for controls (S1; S4; S5; C1); a, b, c, d; *p* > 0.05 represent non-significant statistical differences between samples and controls (Mann–Whitney test by MedCalc program). Legend: A; D—benign prostatic hyperplasia (S1); B; E—nodular hyperplasia, atrophy, and chronic inflammation (S4); C; F—prostate adenoma with chronic inflammation (S5); NS—unstained control prostate cell cultures (S1, S4, and S5); C1—positive control BPH cell cultures.

**Figure 9 ijms-25-09329-f009:**
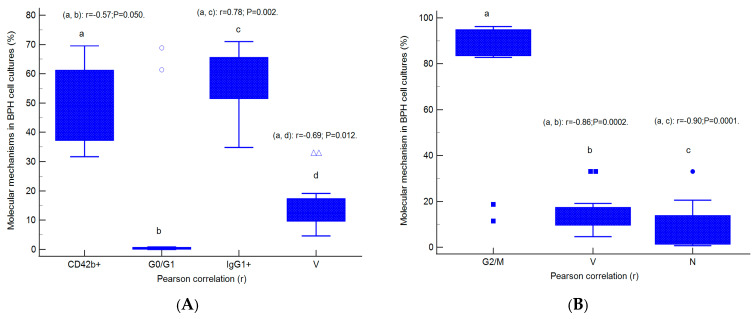
Parametrical correlations (r) between molecular mechanisms in heterogeneous BPH cell cultures with an aggressive phenotype (**A**–**D**). Legend: CD42b+—adhesion glycoprotein positive expression; IgG1+—immunoglobulin positive expression; G0/G1; G2/M; S—cell-cycle phases; V—viability; N—necrosis; EA—early apoptosis; AO+H+—positive cell populations expressed for autophagy and nuclear shrinkage; a, b, c, d; *p* < 0.01 represent significant statistical differences between samples (Pearson test, r, by MedCalc program).

**Figure 10 ijms-25-09329-f010:**
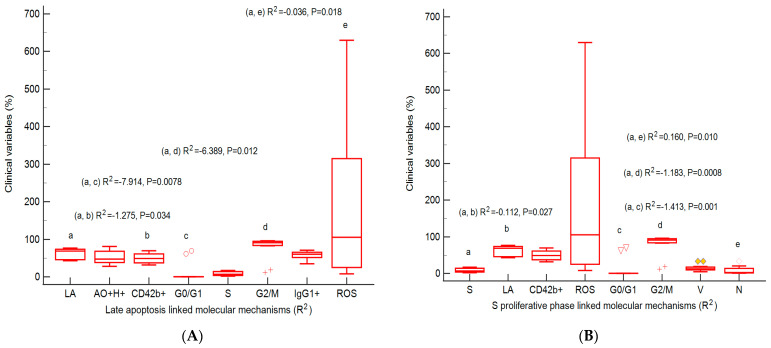
Predictor factors represented by the coefficient of multiple regression (R^2^) for linked molecular mechanisms in heterogeneous aggressive phenotype BHP cell cultures. (**A**) Late apoptosis: AO+H+: R^2^ = 0.180 ± 0.189, *p* = 0.394; CD42b+: R^2^ = −1.275 ± 0.403 *, *p* = 0.034; G0/G1 phase: R^2^ = −7.914 ± 1.603 **, *p* = 0.0078; S phase: R^2^ = −3.422 ± 1.326, *p* = 0.061; G2/M phase: −6.389 ± 1.468 *, *p* = 0.012; IgG1+: R^2^ = 0.135 ± 0.348, *p* = 0.717; ROS: R^2^ = −0.036 ± 0.009 *, *p* = 0.018. (**B**) S proliferative phase of cell cycle: LA: R^2^ = −0.112 ± 0.033 *, *p* = 0.027; CD42b+: R^2^ = −0.099 ± 0.039, *p* = 0.064; ROS: R^2^ = −0.002 ± 0.001, *p* = 0.129; G0/G1 phase: R^2^ = −1.413 ± 0.167 **, *p* = 0.001; G2/M phase: R^2^ = −1.183 ± 0.129 **, *p* = 0.0008; V: R^2^ = 0.078 ± 0.052; *p* = 0.2136; N: R^2^ = 0.160 ± 0.035 **; *p* = 0.010. Legend: a, b, c, d, e; * *p* < 0.05 and ** *p* < 0.01 represent statistically significant differences between variables made by Least squares multiple regression (R^2^) by MedCalc v20.111 Software Ltd., Ostend, Belgium. AO+H+—positive cell populations expressed for autophagy and nuclear shrinkage; CD42b+—adhesion glycoprotein positive expression; G0/G1; G2/M; S—cell-cycle phases; IgG1+—immunoglobulin positive expression; ROS—reactive oxygen species; LA—late apoptosis; V—viability; N—necrosis.

## Data Availability

Data and their interpretation are contained in the research article.

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
