# Peer review of "Apoptosis–Cell Cycle–Autophagy Molecular Mechanisms Network in Heterogeneous Aggressive Phenotype Prostate Hyperplasia Primary Cell Cultures Have a Prognostic Role"

_ijms, 2024, doi:10.3390/ijms25179329_

Round 1

Reviewer 1 Report

Comments and Suggestions for Authors

This study was done and conducted solely by flow cytometry, and it is difficult and not sufficient to explain multiple biological characteristics. Authors are advised to perform additional experiments to support the results. For instance, L124-126, “cell cycle arrest in G2/M phase being characterized by aneuploidy status”. The aneuploidy status requires further evidence to prove.

The results of Fig.2 lack of descriptive explanation.

How would authors explain the BPH cell culture S5 showed a significantly high percentage both at the S phase and G2/M phase?

Comments on the Quality of English Language

Some typing mistakes and words are incorrectly chosen in the manuscript, such as intern control, reported to control.

Author Response

Dear Reviewer 1,

Best regards,

PhD Biologist, Researcher III, Matei Elena

CEDMOG, Ovidius University from Constanta, Romania

Reviewer 2 Report

Comments and Suggestions for Authors

In this manuscript, the authors investigated cell death, DNA content, and autophagy molecular mechanisms network as predictor biomarkers to highlight prostate pathogenesis and its prognostic role. This study is based on flow cytometric analysis.

The authors observed that higher late apoptosis expression, high proliferative S phase of cell cycle category, and aneuploidy status represent unfavorable prognostic biomarkers and dependent predictor factors in heterogeneous aggressive phenotype BHP cell cultures.

Overall, this study is potentially of interest. However, as the results are only based on flow cytometric analysis, different methodology approaches (including WB and IHC) are required to confirm the results and support the conclusions.

Specific comments:

Figure 2 shows nodular hyperplasia, atrophya, and chronic inflammation in primary prostate cell cultures developed from tissue pieces recovered from patients with hyperplasia (BPH). The images are not convincing. Immunohistochemistry (IHC) staining is required using specific markers to study inflammatory cells in the TME.    

Comments on the Quality of English Language

Minor editing of English language is required.

Author Response

Dear Reviewer 2,

Best regards,

PhD Biologist, Researcher III, Matei Elena

CEDMOG, Ovidius University from Constanta, Romania

Round 2

Reviewer 1 Report

Comments and Suggestions for Authors

The manuscript has been revised. No further comments.

Comments on the Quality of English Language

Some typing mistakes in text.

Author Response

(The authors gave the same response as above.)

Reviewer 2 Report

Comments and Suggestions for Authors

Although the authors partially addressed the raised questions, the new version is sufficiently improved.

Author Response

(The authors gave the same response as above.)
